# Effects of Ferric Ions on Cellulose Nanocrystalline-Based Chiral Nematic Film and Its Applications

**DOI:** 10.3390/polym16030399

**Published:** 2024-01-31

**Authors:** Shuaiqi Wang, Bingqun Lin, Yihan Zeng, Mingzhu Pan

**Affiliations:** College of Materials Science and Engineering, Utilization of Forest Resources, Nanjing Forestry University, Nanjing 210037, China; 18356732261@163.com (S.W.); lbq@njfu.edu.cn (B.L.); zengyihan@njfu.edu.cn (Y.Z.)

**Keywords:** chiral nematic materials, cellulose nanocrystal, decreased pitch, ferric ions, liquid crystal

## Abstract

Chiral nematic materials have been attracting attention in fields of advanced functional applications due to their unique iridescent colors and tunable helical structure. A precisely decreased pitch is of importance for construction and applications of chiral nematic materials; however, it remains a huge challenge. Herein, cellulose nanocrystal (CNC) is selected as a constructed matrix for chiral nematic films, and ferric chloride (FeCl_3_) is used as a modification agent. We investigate the effects of the ferric ion loads on the helical structure and optical characteristics of iridescent film. Subsequently, the influence of ferric ions on the assembly process of CNC liquid crystal and the regulation of the structure color of self-assembled monolayers are discussed. Therefore, the CNC/FeCl_3_ chiral nematic films showed a blueshifted structural color from orange to blue, which highlights a simple route to achieve the regulation of decreased pitch. Further, we have applied this CNC/FeCl_3_ chiral nematic film for benzene gas detection. The sensing performance shows that the CNC/FeCl_3_ chiral nematic film reacts to benzene gas, which can be merged into the nematic layer of the CNC and trigger the iron ions chelated on the CNC, consequently arousing the redshift of the reflected wavelength and the effective colorimetric transition. This CNC/FeCl_3_ chiral nematic film is anticipated to boost a new gas sensing mechanism for faster and more effective in-situ qualitative investigations.

## 1. Introduction

Chiral nematic materials are attracting attention in fields of intelligent sensors, decoration, anti-counterfeiting, and encryption due to their unique iridescent colors and tunable helical structures [1,2,3,4]. What creates iridescent colors, also known as structural colors, is a fine structure with a spatially ordered lattice, which exhibits brilliant colors under refraction, diffuse reflection, diffraction, or interference of light [5,6,7,8]. Interestingly, this iridescent color has been discovered in natural creatures, such as beetles, butterflies, and fish. Inspired by nature, great efforts have been put forward on the fabrication of chiral nematic materials with a crystalline TiO_2_ microsphere [9], thiol-acrylate chemistries [10], and semi-interpenetrating polymer network [11].

Cellulose, one of the important biopolymers, is the richest biomass in the world [12]. Cellulose nanocrystals (CNCs) from sulfuric acid hydrolytic cellulose are a kind of natural photonic crystal with negatively charged groups on its surface. They can form a chiral nematic film with left-handed helical structure through self-assembly induced by the evaporation of water. The resultant chiral nematic film shows extraordinary optical properties and provides an iridescent color structure [13,14]. The unique optical characteristics of CNC chiral liquid crystals make it possible to develop in the field of colorimetric gas sensors. In addition, there are abundant active groups, such as hydroxyl and carboxyl, within cellulose molecules. On the other hand, they can automatically respond to humidity, aldehydes, or alcohol gas molecules by forming hydrogen bonds, so as to achieve the transformation of bright colors [15,16]. On the other hand, it can also be used as an excellent functional template carrier to achieve higher response characteristics or can be modified to give it special functions [17,18,19]. Up to now, fabrication of CNC-based chiral nematic materials with a precise helical structure remains a huge challenge.

To achieve the precise helical structure within chiral nematic materials, many modification strategies have been put forward: (1) regulating sulfuric acid hydrolytic conditions, such as acid-to-pulp ratio and the hydrolysis temperature [20]; (2) incorporation of metal ion, such as Cu^2+^, Na^+^, Al^3+^, or K^+^ [16,21,22,23]; (3) incorporation of small molecules, such as glucan [24]; (4) incorporation of neutral polymers, such as polyethylene glycol (PEG) [25] or waterborne polyurethane (PU) [26]; (5) incorporation of polyelectrolytes, such as polyacrylic acid (PAA) [3]; and (6) applying an ultrasmall magnetic field [27]. These modifications have effectively regulated the helical structure and mechanical properties of chiral nematic materials. However, the pitch of chiral nematic materials is almost always increased, which is not beneficial for advanced functional development. Limited studies have reported the modification methods for a decreased pitch 2127, which is of importance for chiral nematic materials.

In this study, CNC is selected as a constructed matrix for chiral nematic films, and the ferric chloride (FeCl_3_) is used for a modification agent. We investigate the effects of the ferric ion loads on the helical structure and optical characteristics of iridescent film. Subsequently, the influence of ferric ions on the assembly process of CNC liquid crystal and the regulation of the structure color of self-assembled monolayers are discussed. Therefore, the CNC/FeCl_3_ chiral nematic films showed blueshifted structural color from orange to blue, which provides a potential strategy to precisely regulate the pitch of chiral nematic materials.

## 2. Experimental Section

### 2.1. Materials

Microcrystalline cellulose (MCC) was provided by Shanhe Pharmaceutical Excipients Co., Ltd. (Huainan, China). H_2_SO_4_ (95–98 wt%) and ferric chloride (FeCl_3_·6H_2_O, 99.5 wt%) were supplied from Sinopharm Chemical Reagent Co., Ltd. (Nanjing, China). The detailed characterization methods are provided in the Appendix A.

### 2.2. Preparation of CNC/FeCl_3_ Colloids

A typical preparation of CNC/FeCl_3_ chiral nematic film is described as follows: CNC colloid with a concentration of 2.0 wt% was fabricated via 64 wt% sulfuric acid hydrolysis according to our previous work 20, and the detailed preparation of CNC is illustrated in the Appendix A. The resultant CNC colloid has a zeta potential value of −43.2 mV and OSO_3_- content of 0.32%. Subsequently, 100 μL FeCl_3_ aqueous solution (2, 4, 6, 8, 10, 12, 14, 16, and 18 mmol/L) was added to 3 mL CNC suspension; wherein, the corresponding FeCl_3_ load is 32, 63, 94, 126, 157, 188, 220, 252, and 283 mmol/(g·CNC). The mixture was mildly stirred for 5 min; finally, a CNC/FeCl_3_ colloidal suspension was prepared.

The obtained hybrid was subsequently placed in polystyrene Petri dishes (~30 mm in diameter). Subsequently, CNC/FeCl_3_ chiral nematic films were formed by evaporation-induced self-assembly (EISA) under ambient conditions for three days, and then, all films were conditioned at 20 °C and a relatively humidity of 40 ± 5%. The as-obtained films have typical thicknesses of approximately 30–70 μm, as shown in Figure 1, and are denoted as CFx, where C and F stand for CNC and FeCl_3_, respectively, and x refers to different loadings of FeCl_3_. For example, CF_2_ represents 2 mmol/L FeCl_3_. For comparison, the CNC without FeCl_3_ chiral nematic film was also prepared. Accordingly, under natural light, these obtained CNC and CNC/Fe films exhibited iridescent color, corresponding to orange (CNC), bright orange (CF2), greenish yellow (CF4), mint green (CF6), cyan (CF8), turquoise (CF10), sky blue (CF14), and transparent (CF16, CF18).

## 3. Results and Discussion

### 3.1. Morphological Characteristics of CNC/FeCl_3_ Colloids and Chiral Nematic Films

Figure 2 illustrates the morphological images of CNC and CNC/FeCl_3_ colloids. Neat CNC was oriented horizontally with a typical rod-like crystal that was 17.8 ± 2.1 nm in diameter and 300 ± 25 nm in length. Comparatively, the introduction of FeCl_3_ induced CNC aggregation. Specially, the tactoid aggregation and conformation can be precisely regulated with the FeCl_3_ concentration, which will be discussed later. The cross-sectional images of CNC/FeCl_3_ films showed a uniform arrangement, with apparent periodic spacing and spiral stacking, as shown in Figure 3. The pitch of neat CNC chiral nematic film was 382 nm, and the pitch of CNC/FeCl_3_ films decreased to 329 nm for CF2. Moreover, CF6 and CF10 were also helically arranged in layers with periodic spacing, and the pitch gradually decreased to 261 nm and 238 nm. However, the helically arranged structure was not clear for CF14, and it disappeared for CF18.

The morphologies of the prepared CNC and CF6 films were further studied with AFM measurement. Compared to the morphology of CNC film, the CF6 films had a more compact structure with nano-sized CNCs densely packed on the substrate. As illustrated in Figure 4a,b, neat CNC film had Ra roughness ≈1.45 nm, which revealed a formation of rough CNC films with low CNC number density. Incorporation of 6 mmol/L FeCl_3_ resulted in a smooth CF6 film with a reduced Ra roughness of ≈0.78 nm, which indicated a formation of densely packed CNC films. Here, we use the center-to-center distance between adjacent CNCs as a parameter to scale the packing density [28]. As shown in Figure 4(a_2_,a_3_), the neat CNC film formed a dense stacking structure with an average center-to-center distance of 17.25 nm. Moreover, CNC/FeCl_3_ chiral nematic films with 6 mmol/L FeCl_3_ had a decreased packing density with a center-to-center distance of 16.13 nm, as shown in Figure 4(b_2_,b_3_). This can also be confirmed from their phase diagrams (Figure 4(a_1_,b_1_)).

### 3.2. Optical Properties of CNC/FeCl_3_ Chiral Nematic Films

Optical properties of CNC and CNC/FeCl_3_ chiral nematic films were performed with UV–Vis spectra and CD spectra. Clearly, the CNC chiral nematic film exhibited the maximal reflection wavelengths (*λ*_max_) at 671 nm in Figure 5a,b. With the incorporation of FeCl_3_, the reflection wavelengths gradually decreased at 636 nm (CF2), 603 nm (CF4), 558 nm (CF6), 536 nm (CF8), 526 nm (CF10), 484 (CF12), and 442 nm (CF14), which displayed an obvious blueshifted phenomenon. What is important is that the chiral nematic structure disappeared for CF18 due to phase separation. To reveal the influence of FeCl_3_ on the iridescent films, the UV–Vis absorbance spectra were transformed into Commission on Illumination (CIE) chromaticity values (Figure 5c). They were in the sequence orange (CNC), bright orange (CF2), greenish yellow (CF4), mint green (CF6), cyan (CF8), turquoise (CF10), and sky blue (CF14). It is observed that CNC/FeCl_3_ chiral nematic structure successfully remained in resultant films and was precisely regulated by FeCl_3_ content. Furthermore, CD spectra also showed a gradual blueshift in λ_max,_ which peaked at the visible region (Figure 5d) with the increase in FeCl_3_ from 0 to 14 mmol/L. The presence of a strong positive signal in the CD spectra indicates a left-handed chiral nematic organization in CNC/Fe films [29].

### 3.3. Chiral Nematic Formation of CNC/FeCl_3_ Iridescent Film

In order to compressively clarify the regulation of the chiral nematic phase of CNC/FeCl_3_ suspension correlated with FeCl_3_ content, POM was used to monitor the fingerprint textures and tactoid evolution during EISA. As depicted in Figure 6, the anisotropic phase was first formed at the early stage of the EISA process due to electrostatic repulsion and Coulombic attraction. Then, the anisotropic phase gradually grew and formed a liquid crystal phase with a birefringence phenomenon. As the evaporation continued, the liquid crystal phases contacted each other and then formed a fingerprint texture with a uniform pitch [3].

On the other hand, tactoids are first formed in the isotropic phase, inducing a phase separation relationship between isotropic and anisotropic phases in CNC suspensions. Subsequently, the tactoids are coalesced and deposited on the bottom of the sample, forming a high-density liquid crystal phase [23,30]. We can find that by adding FeCl_3_ to the suspension, the size of the tactoids decreases significantly. However, no obvious liquid crystal phase structure was observed in CF18. This is because excessive iron ions will greatly reduce the electrochemical repulsion on the surface of the CNC, and CNC agglomerates together to inhibit the phase transition process. It can be found that with the addition of FeCl_3_, the charge interaction between FeCl_3_ and CNC and the coordination between FeCl_3_ and hydroxyl are not obvious at low concentrations. With increasing FeCl_3_, metallic ferric ions were attracted by the chelation action sourced from multi-hydroxyl groups as well as the electrical affinity of negatively charged sulfate bearing on the surface of CNCs, thereby forming electrical ionic double layers as reported in the event of electrolyte effects, which makes the pitch of the film smaller [16]. When the concentration of FeCl_3_ continues to increase, the charge mutual attraction and coordination to hydroxyl will further increase, exceeding the repulsion force forming the pitch, and finally leading to the disappearance of the pitch. This is consistent with the results of the observed color change.

The zeta potential value of neat CNC (−43.2 mV, Figure 7a) is typical of polyanion species. With the incorporation of FeCl_3_, the zeta potential became −42.3 mV (CF2) and −33.4 mV (CF14). A decreased net electrostatic charge was taken as an indication of self-assembly that resulted from the electrostatic attraction between the anionic groups (OSO_3_^−^ in CNC) and FeCl_3_ [3]. With 18 mmol/L FeCl_3_, the enhanced electrostatic attraction, as well as the coordination interaction between ferric ions and OH^−^ in CNC molecules, led to a decreased absolute zeta potential (−28.6 mV for CF18) and iridescence disappearance, which is in accordance with Coulomb’s law (coulombic attraction) [31]. Further, neat CNC film had the absorption peak of hydroxyl groups at 3267 cm^−1^ in FTIR spectra. Whereas it shifted to 3338 cm^−1^ (CF2) and then to 3266 cm^−1^ (CF18), as is indicative of a hydrogen bonding formation between CNC and FeCl_3_ (Figure 7b). Additionally, the Fe-O bond at about 720 cm^−1^ is present for CF14 and CF18, and the coordination interaction between FeCl_3_ and hydroxyl groups 2800–2900 cm^−1^ is also observed [32]. The electrostatic repulsion, hydrogen bonding, and coordinated interaction simultaneously regulated the CNC/FeCl_3_ chiral nematic films during the self-assembly process (Figure 7c). It is noted that the presence of FeCl_3_ gives rise to electrostatic attraction between the OSO_3_^−^ and ferric ions. At a lower FeCl_3_ concentration, CNCs can form and assemble at the interface due to the strong electrostatic repulsion, inducing the densely packed structures. With increasing FeCl_3_ concentration, the number of free protons from the sulfate groups and hydroxyl decrease, leading to electrostatic attraction and more densely packed CNC assemblies [28,33].

### 3.4. Applications for Aromatic Hydrocarbon Response

As the chemical industry develops rapidly, the quality of air, which is closely related to human life, decreases dramatically with the emission of aromatic hydrocarbon vapors. When the prepared CNC/FeCl_3_ chiral nematic film was placed in a self-made reaction chamber to study the sensing response of saturated benzene vapor, the color change before and after CF6 film was the most obvious (Figure 8a,b). Therefore, we chose to use a photonic crystal sensor assembled with blue-green CF6 to detect aromatic hydrocarbon vapors at different concentrations.

Benzene vapor was first tested to capture the colorimetric response by recording the structure color and measuring the reflectance spectrum. It was observed that the concentration of benzene vapor gradually increased from 0.00 to 1000 g/m^3^, the color of the structure changed from the initial blue-green to the final black-red (Figure 9a), and the position of the stop band correspondingly changed from 558 nm to 663 nm (Figure 9b). The same trend was also found in the CD spectra of the films after response under corresponding conditions (Figure 9c). It shows that after benzene gas is adsorbed by the film, the pitch of the film changes, resulting in a change in film color. An increase in vapor concentration leads to a change in the refractive index of the CNC/FeCl_3_ film layer, which alters the optical properties of the sensor and produces a significant redshift in the reflectance spectrum, which agrees well with the predictions of Bragg’s law. In addition, it was found that the rate of change of the maximum reflected wavelength decreased at concentrations above 500 g/m^3^, as benzene occupied all active sites on the film. After the response of the CNC film in the benzene vapor environment, the color and maximum reflection wavelength of the film remained basically unchanged, indicating that pure CNC will not adsorb benzene vapor, which further confirms that ferric ions play a sensitive role in CF6 film.

To evaluate the selectivity, the CNC colorimetric sensor was exposed to other VOCs. Moreover, wavelength diagrams of this sensor for sensing various concentrations of toluene, xylene, and acetone vapors with the correlation coefficients were also given in Figure 9e. It can be found that the CNC/FeCl_3_ film shows excellent sensitivity to aromatic gases. Moreover, the neat CNC films do not show an obvious color change in the benzene vapor environment with different concentrations (Figure 9d), which indicates that the color change of CF6 films in benzene vapor is caused by ferric ions. It should be noted that each gas resulted in a different color change wavelength. For aromatic hydrocarbon vapor, the more the number of methyl groups on the benzene ring, the stronger the molecular activity, and the stronger the adsorption with ferric ions. Therefore, under the same concentration of vapor environment, CF6 film shows higher sensitivity to xylene gas (Figure 9f). An evaluation of the reversible transition of the stop band position of the sensor for alternating exposure to 0 and 400 g/m^3^ benzene, as shown in Figure 9g, indicated that the sensor exhibited very good reversibility and repeatability over eight cycles of testing.

AFM measurements of CNC and CF6 films with different benzene vapor concentrations showed that the surface roughness of the films after gas sensitive adsorption was improved to varying degrees (Figure 10). It can be found that the surface morphology of neat CNC films in the benzene vapor environment will not change significantly (Figure 10a–a_2_). However, when the benzene vapor concentration reaches 400 g/m^3^, the CF6 surface has a convex envelope structure due to the swelling of the gas (Figure 10b–b_2_. The increase in vapor concentration leads to an expansion of the CNC/FeCl_3_ layer and an increase in the layer thickness, which causes a change in the optical properties of the sensor and produces a significant redshifted reflectance spectrum. In order to better prove that benzene molecules are indeed adsorbed by CF6 films, we also tested the FTIR of CF6 films before and after response. It can be seen from Figure 8c that in the spectrum of the sample after the response, new characteristic peaks belonging to benzene ring appear at 1455 cm^−1^, 2333 cm^−1^, and 2930 cm^−1^, and the peak intensity increases to varying degrees with the increase in vapor concentration.

As illustrated, although adsorption capacity of benzene gas onto CNC is limited, CNCs with certain ferric-ion-loading did change the possibility because of the strong chelation affinity of ferric ions to benzene gas. This may be due to the acid-based chemical interaction between the CNC/FeCl_3_ composite film and the benzene ring due to the existence of ferric ions, which makes the benzene vapor adsorb in the composite film and changes the pitch of the film. Delocalized π electrons of aromatic rings can pass through unfilled 3d metal orbitals with transition metal cations σ bond sharing electron density [34,35]. Because of the strong chelating affinity of iron ions for benzene gas, benzene gas will be attracted by the iron ions located between the CNC layers, which causes the expansion of the neighboring CNC layers, leading to an increase in the *p* value. However, the cholesteric mesophase of CNC was destroyed after the overdoping of CNC with 18 mmol/L of iron ions, and even though the iron ions could trap benzene, they had no significant effect on the very few mesophases in CNC that produced the color of the thin film structure. Therefore, it is not difficult to imagine that the perfect cholesteric structure of CNC and the good sensitive triggering of ferric ions have a crucial influence on the colorimetric sensing (redshift). In other words, as a benzene sensor, the separate film of CF6 can cause a significant reflection wavelength shift and a visible colorimetric change.

It is known that CNC-based composite films can interact with water through hydrogen bonding. Since water is required for the gating action of benzene gas on the moisture sensitivity of CNC films, the effect of benzene on the moisture sensitivity of CNC films and the effect of ambient relative humidity on the benzene sensitivity of CNC/FeCl_3_ films were further investigated. Figure 11 shows the results of the humidity response of CNC/FeCl_3_ iridescent films in gaseous benzene with *C_b_* of 100, 500, and 750 g/m^3^. The *λ*_max_ values of this films were found at 567, 575, 598, 610, and 626 nm at a relative humidity of 44, 54, 76, 87, and 99% (at 100 g/m^3^), respectively (Figure 11a). The *λ*_max_ value of the CNC/FeCl_3_ iridescent film was redshifted by 59 nm at 100 g/m^3^ of *C_b_*. As expected, the color of the CNC/FeCl_3_ iridescent film observed by the naked eye was also redshifted with the increase in RH at 100 g/m^3^, as shown in Figure 11a. These results indicate that the humidity sensitivity of CNC/FeCl_3_ iridescent films can be controlled by exposing the films to benzene gas. Similarly, the humidity response of CNC/FeCl_3_ films ranges from 46 nm (Figure 11b) to 35 nm (Figure 11c) when *C_b_* is 500 and 750 g/m^3^, respectively. All these results show that the λ_max_ of the CNC pentachromatic film increases and the color undergoes a significant redshift due to the steric contribution of the benzene molecules under different *C_b_* conditions.

In order to further understand the induction mechanism of benzene, we use Gaussian 09 as a tool to simulate the adsorption energies of C_6_H_6_ molecules and H_2_O molecules on top of ferric ions and CNC. The former is based on the σ-bonding of the shared electron density between benzene and iron ions. The latter is due to the hydrogen bonding interaction between H_2_O and CNC. Therefore, exploring whether the ferric ions in CF6 films play a key role in the adsorption of benzene is interesting for further material design. However, this goal is difficult to achieve experimentally, so we used Gaussian 09 software to calculate the reaction energy barrier (Δ*E*) values by simulating the adsorption of benzene and water molecules on different sites of CNC. Through the enthalpy of the reaction between CNC and benzene before and after the introduction of ferric ions, the priority of reaction is judged, and the outstanding contribution of iron ions in the adsorption of benzene gas is confirmed.

In order to reduce the calculation cost, we simplified the cellulose chain as a monomer to participate in the calculation. In the optimization calculation, the system of CNC and benzene cannot be in a stable state, which shows that CNC and benzene will not react spontaneously (Figure 12a). The reaction energy barrier of CNC containing ferric ions with benzene vapor is Δ*E* = −21.78 kJ.mol^−1^, which shows that the adsorption of benzene by CNC/FeCl_3_ system is a spontaneous reaction, which is also consistent with the experimental results (Figure 12d). As expected, the calculated result Δ*E* > 0 between CNC and benzene molecules shows that CNC really cannot adsorb benzene vapor (Figure 12b). However, it is surprising that in the CNC/FeCl_3_ system, the simulation results show that the free energy of ferric ions and the water binding system is much more than the sum of their separate free energy, and during the optimization process, water molecules have the tendency to form hydrogen bonds with the carboxyl group on the edge of ferric ions. It shows that in the adsorption process, the hydrogen bonding force between H_2_O molecules and CNC is much greater than that between water molecules and ferric ions.

Notably, the humidity response range gradually decreased with increasing C_b_. This is because there are abundant hydrophilic groups on the surface of CNC, which is easy for water molecules to bind, and ferric ions have excellent hydration, which means that water molecules may seize the active sites in ferric ions [36,37]. Therefore, there is a competitive relationship between benzene vapor and water molecules, and at the same time, they have a synergistic effect on the redshift of the film. It can be found from Figure 13 that with the increase in benzene vapor concentration, the response range of the CNC/FeCl_3_ film to ambient humidity gradually decreases, which may be caused by the presence of high concentration benzene vapor inhibiting the hydrogen bond binding force between water molecules and the film, likely because of the formation of σ bond between benzene molecules and ferric ions, leading to reduced numbers of humidity response sites.

## 4. Conclusions

In this study, a CNC/FeCl_3_ chiral nematic film was successfully fabricated. Among samples, CNC is selected as a constructed matrix and the ferric chloride (FeCl_3_) is used as a modification agent. We investigated the effects of the ferric ion loads on the helical structure and optical characteristics of iridescent film. With an appropriate content of ferric ions, chiral nematic films exhibited a progressive decrease in *λ*_max,_ from 671 nm (neat CNC) to 442 nm (CF14). It is indicative of a maximum blueshifted wavelength of 229 nm because of the strong electrostatic attraction between OSO_3_^−^ and ferric ions. Subsequently, the influence of ferric ions on the assembly process of CNC liquid crystal and the regulation of the structure color of self-assembled monolayers were discussed. Therefore, the CNC/FeCl_3_ chiral nematic films showed blueshifted structural color from orange to blue, which provides a potential strategy to precisely regulate the pitch of chiral nematic materials. Finally, we used the CNC/FeCl_3_ chiral nematic films for aromatic hydrocarbon gas detection. This CNC/FeCl_3_ chiral nematic film displayed a degree of the sensing performance, and the sensitive sensing for aromatic hydrocarbon gas detection will be emphasized in future research.

## Figures and Tables

**Figure 1 polymers-16-00399-f001:**
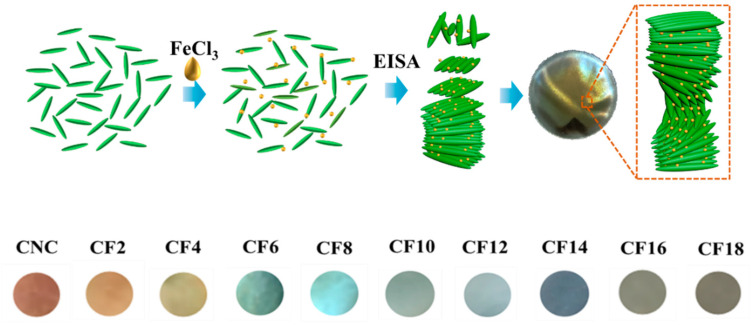
Preparation of the CNC/FeCl_3_ chiral nematic film and its structural relevant color.

**Figure 2 polymers-16-00399-f002:**
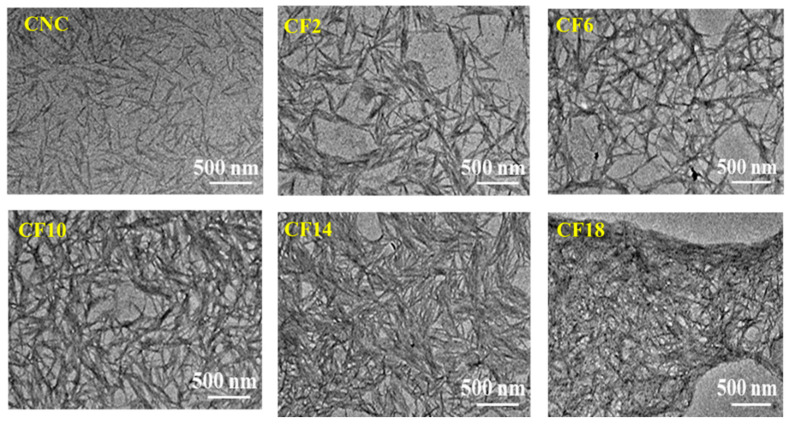
Morphological images of CNC/FeCl_3_ colloids from TEM measurement.

**Figure 3 polymers-16-00399-f003:**
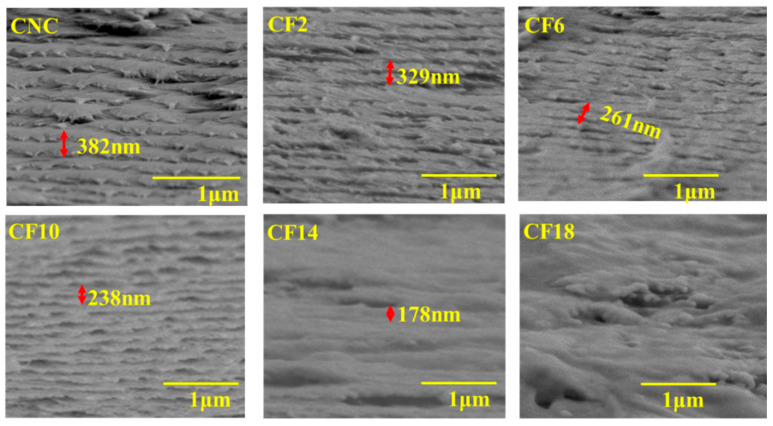
Morphological images of CNC/FeCl_3_ chiral nematic films from FE-SEM measurement.

**Figure 4 polymers-16-00399-f004:**
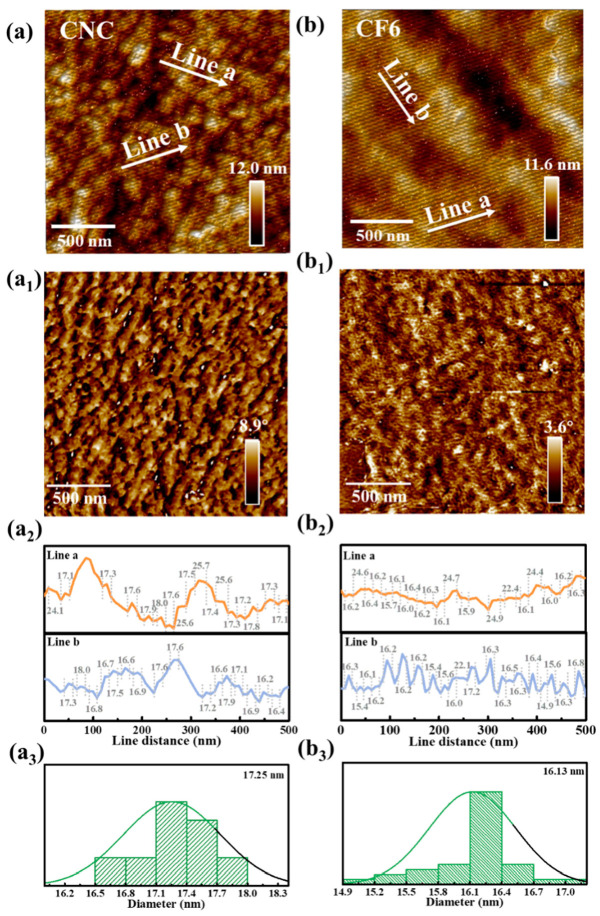
AFM topography of films with CNC and CF6: (**a**,**b**) height images; (**a_1_**,**b_1_**) phase images; (**a_2_**,**b_2_**) show the line profiles of the fiber widths obtained on the CNC and CF6 surfaces; (**a_3_**,**b_3_**) represent the distribution of fiber width.

**Figure 5 polymers-16-00399-f005:**
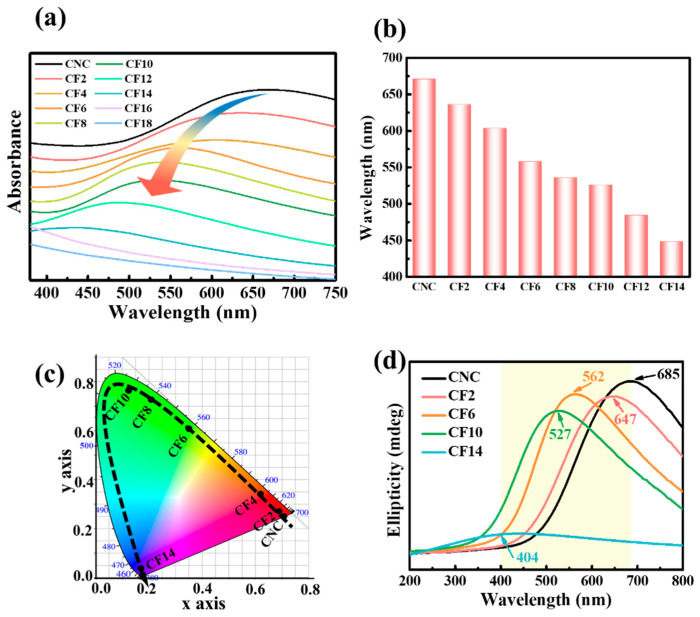
Optical characteristics of CNC/FeCl_3_ chiral nematic films. (**a**,**b**) UV–Vis spectra, (**c**) CIE chromaticity diagram, and (**d**) CD spectra.

**Figure 6 polymers-16-00399-f006:**
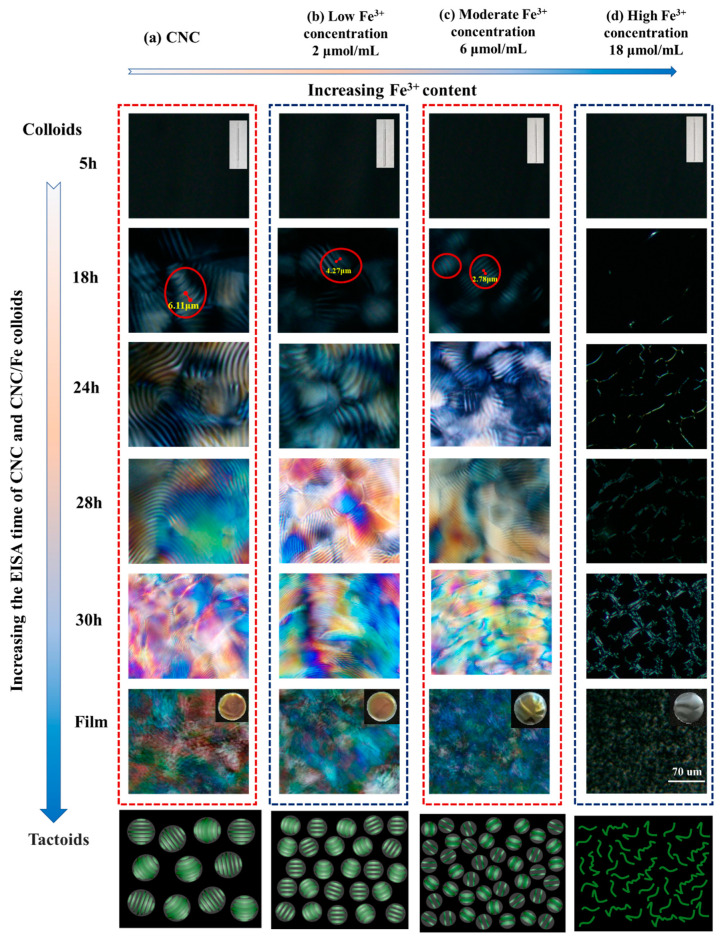
Evolution of the fingerprint and tactoid conformation correlated with FeCl_3_ content during EISA of CNC/Fe using POM measurement.

**Figure 7 polymers-16-00399-f007:**
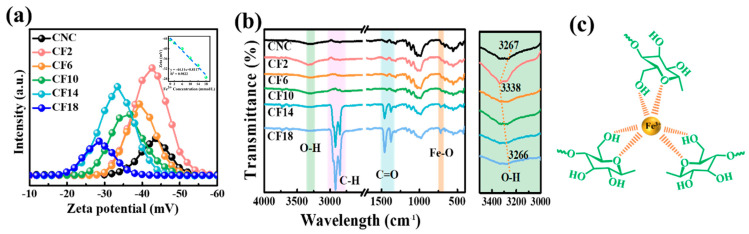
(**a**) Zeta potential value of CNC/FeCl_3_ complexes; the inset image represents the correlated relationship between FeCl_3_ and zeta potential value. (**b**) FTIR spectra of CNC/FeCl_3_ chiral nematic films. (**c**) The interaction bonds between CNC and FeCl_3_.

**Figure 8 polymers-16-00399-f008:**
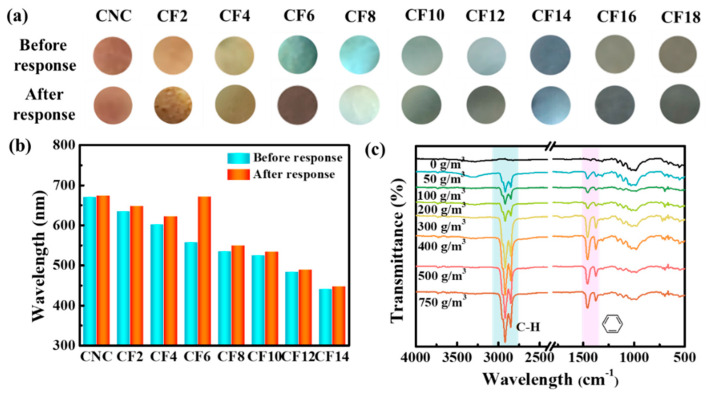
Response of CNC/FeCl_3_ films to saturated benzene vapor: (**a**) physical drawing, (**b**) UV–Vis data change diagram; (**c**) FTIR data diagram of CNC/FeCl_3_ films after response.

**Figure 9 polymers-16-00399-f009:**
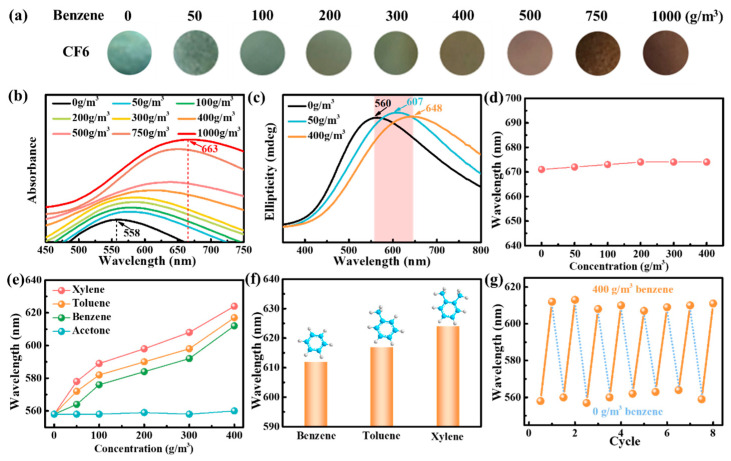
Benzene-responsive results of the CF6 photonic crystal film: (**a**) digital images; (**b**) UV–vis absorbance spectra; (**c**) CD spectra; (**d**) *λ*_max_ change of neat CNC films in response to benzene vapor; (**e**) gas sensitivity of CF6 to different gas; (**f**) maximum reflection wavelength change of CF6 in different aromatic hydrocarbon vapors of 400 g/m^3^; (**g**) cycle performance of CF6.

**Figure 10 polymers-16-00399-f010:**
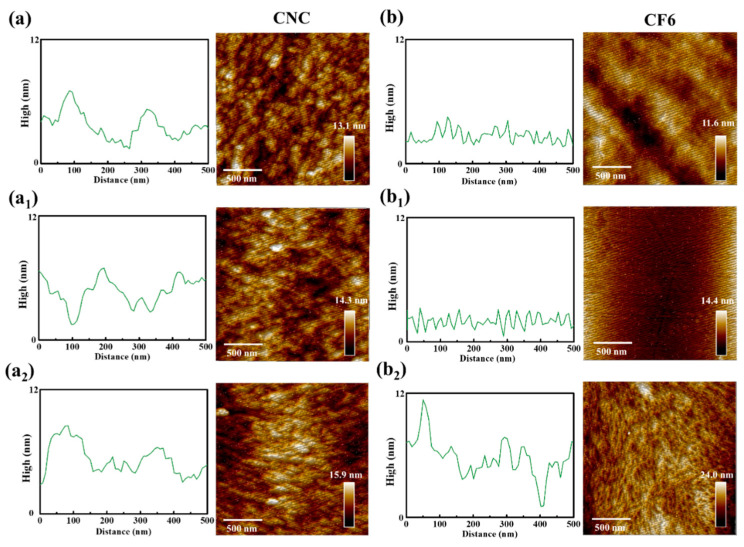
Surface AFM diagram of CNC and CF6 after response to different benzene vapors: (**a**,**b**) 0 g/m^3^; (**a_1_**,**b_1_**) 50 g/m^3^; (**a_2_**,**b_2_**) 400 g/m^3^.

**Figure 11 polymers-16-00399-f011:**
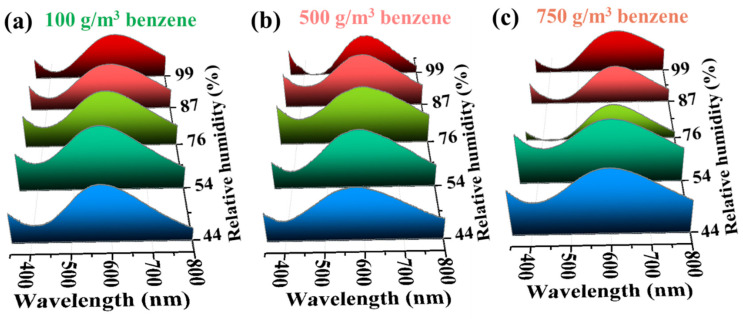
Humidity-responsive results of the CNC/FeCl_3_ film when exposed to gaseous benzene with *C*_b_ of (**a**) 100 g/m^3^, (**b**) 500 g/m^3^, and (**c**) 750 g/m^3^.

**Figure 12 polymers-16-00399-f012:**
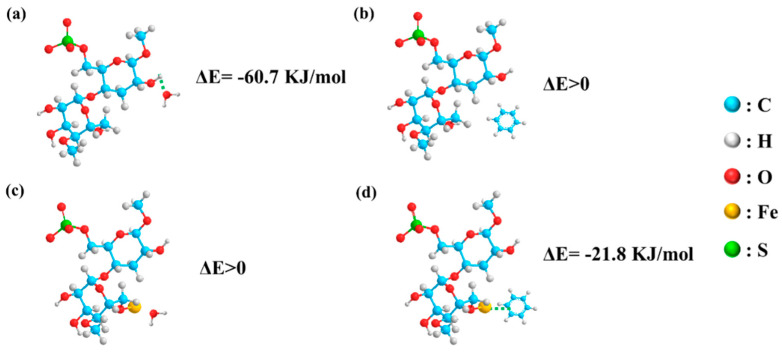
DFT calculation of reaction energy barriers Δ*E* of different systems: (**a**) CNC and H_2_O; (**b**) CNC and benzene; (**c**) CNC/FeCl_3_ and H_2_O; (**d**) CNC/FeCl_3_ and benzene.

**Figure 13 polymers-16-00399-f013:**
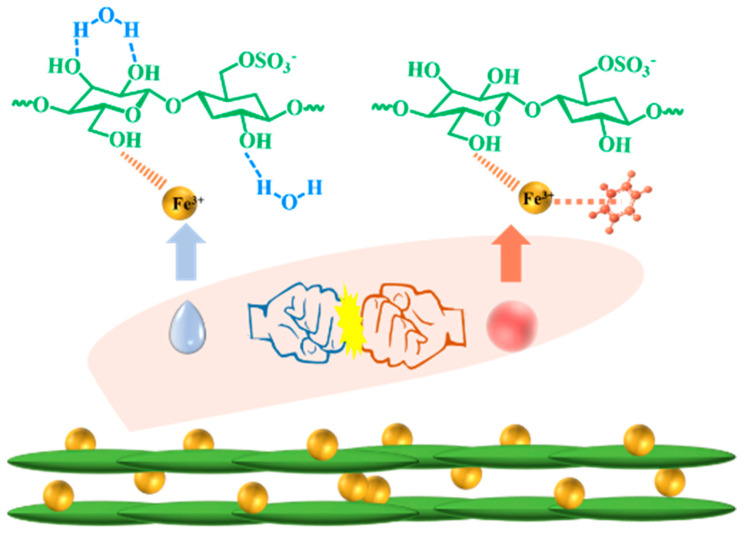
Schematic diagram of competitive adsorption mechanism of benzene and water molecules.

## Data Availability

Data are contained within the article.

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
