# Peer review of "Effects of Ferric Ions on Cellulose Nanocrystalline-Based Chiral Nematic Film and Its Applications"

_polymers, 2024, doi:10.3390/polym16030399_

Round 1

Reviewer 1 Report

Comments and Suggestions for Authors

The authors reported on the preparation of chiral nematic liquid crystal films of cellulose nanocrystals doped with ferric ions and their sensing benzene vapor.  This article is of interest to me but I am not sure of its general appeal in the broad and heterogeneous readership of “Polymers”.  I personally would have given preference to submission to a more specialised journal.  When the editors decide to accept this manuscript for publication in “Polymers”, it is then advised to adapt the following changes:

[1] The authors already published the reports relating to the chiral nematic liquid crystal films of cellulose nanocrystals. The authors should describe the different and appealing points of this report rather than the authors previous reports.  In addition, the strategy of this work is similar to the previous report by D. Liu et al. (Carbohydr. Polym, 2017, 174, 531-539).  Otherwise, this report seems duplicate submission.

[2] In this manuscript, Reference [16] is same as Reference [19].  The authors should check carefully your manuscript before submission.

[3] I think that the English language needs major corrections for the readers.  For example, “Furtherly” in line 17 of page 1 is mistake.  Checking of the English language must be undertaken by negative speakers. 

Author Response

Comments and Suggestions 1:

The authors reported on the preparation of chiral nematic liquid crystal films of cellulose nanocrystals doped with ferric ions and their sensing benzene vapor.  This article is of interest to me but I am not sure of its general appeal in the broad and heterogeneous readership of “Polymers”.  I personally would have given preference to submission to a more specialised journal.  When the editors decide to accept this manuscript for publication in “Polymers”, it is then advised to adapt the following changes:

1) The authors already published the reports relating to the chiral nematic liquid crystal films of cellulose nanocrystals. The authors should describe the different and appealing points of this report rather than the authors previous reports.  In addition, the strategy of this work is similar to the previous report by D. Liu et al. (Carbohydr. Polym, 2017, 174, 531-539). Otherwise, this report seems duplicate submission.

Response: We review the previous report by D. Liu et al. (Carbohydr. Polym, 2017, 174, 531-539). This report designed an ammonia gas sensor based on cholesteric liquid crystal films of copper(II)-doped cellulose nanocrystals (CNC Cu(II)) whose structure, optical and sensing properties were investigated. Here, we prepare CNC/FeCl3 films for benzene gas detection, wherein, the unique ring structure of benzene makes it have high chemical activation energy and chemical stability compared to ammonia gas. Therefore, how to detect aromatic hydrocarbon vapor efficiently has always been a problem to be solved. We introduce FeCl3 into CNC films, and investigate the interactions between the CNC/FeCl3 films and benzene.

2) In this manuscript, Reference [16] is same as Reference [19].  The authors should check carefully your manuscript before submission.

Response: We regret for this misquote, and we revised it.

[16] Dai, S.D.; Prempeh, N.; Liu, D.G.; Fan, Y.M.; Gu, M.Y.; Chang, Y. Cholesteric film of Cu(II)-doped cellulose colorimetric sensing of ammonia gas. Carbohydr. Polym, 2017, 174: 531-539.

[19] He, Y.D.; Zhang, Z.L.; Xue, J.; Wang, X.H.; Song, F.; Wang, X.L.; Zhu, L.L.; Wang, Y.Z. Biomimetic Optical Cellulose Nanocrystal Films with Controllable Iridescent Color and Environmental Stimuli-Responsive Chromism. ACS Appl. Mater. Interfaces 2018, 10, 6, 5805–5811

3) I think that the English language needs major corrections for the readers.  For example, “Furtherly” in line 17 of page 1 is mistake.  Checking of the English language must be undertaken by negative speakers.

Response: We regret for the description, and we revised it.

Reviewer 2 Report

Comments and Suggestions for Authors

1. Fig. 4 would it be appropriate to call :"cellulose crystal length" instead of fiber length? I do not see fibers.

2. Fig. 13 Please annotate benzene 

3. Please elaborate on mechanism on on benzene is absorbed on CNC treated with FeCl3. 

4. Are the films made with CNC and FeCl3 brittle? How long does it take for evaporation method to form film?

5.  Did the authors test any dispersion or sonication of MCC/CNC prior to addition of FeCL3?

6. Please add details on how was MCC converted to CNC?

Author Response

Comments and Suggestions 2:

1). Fig. 4 would it be appropriate to call :"cellulose crystal length" instead of fiber length? I do not see fibers.

Response: Thanks for all valuable suggestion. We regret for the morphological results, and we used “cellulose crystal length” instead of fiber length.

2). Fig. 13 Please annotate benzene

Response: Thanks for all valuable suggestion. We have annotated benzene in Figure 13.

3). Please elaborate on mechanism on benzene is absorbed on CNC treated with FeCl3.

Response: In order to understand the absorption mechanism of benzene on CNC/FeCl3, we use AFM and FTIR to investigate the absorption process of benzene in Figure 8c and Figure 10. As illustrated, although adsorption capacity of benzene gas onto CNC is limited, CNCs with certain ferric-ion-loading did change the possibility because of the strong chelation affinity of ferric ions to benzene gas. It may be due to the acid-based chemical interaction between the CNC/FeCl3 composite film and the benzene ring due to the existence of ferric ions, which makes the benzene vapor adsorb in the composite film and changes the pitch of the film. Delocalized π electrons of aromatic rings can pass through unfilled 3d metal orbitals with transition metal cations σ bond sharing electron density[34] [35]. Because of the strong chelating affinity of iron ions for benzene gas, benzene gas will be attracted by the iron ions located between the CNC layers, which causes the expansion of the neighboring CNC layers, leading to an increase in the P value, which causes a change in the optical properties of the sensor and produces a significant red-shifted reflectance spectrum.

4). Are the films made with CNC and FeCl3 brittle? How long does it take for evaporation method to form film?

Response: Yes, the CNC/FeCl3 films exhibit brittle, and we will conduct much work to improve the toughness in the future work. CNC/FeCl3 chiral nematic films were formed by evaporation-induced self-assembly (EISA) under ambient conditions for three days.

5). Did the authors test any dispersion or sonication of MCC/CNC prior to addition of FeCL3?

Response: Yes. The colloidal solution of CNC was dispersed by ultrasound treatment prior to addition of FeCl3

6). Please add details on how was MCC converted to CNC?

Response:15 g of MCC was hydrolyzed for 50 min for preparation of the suspension. Hydrolysis was performed with 64 wt. % sulfuric acid at acid-to-pulp ratio of 8:1 at 45 °C and 60 °C. The suspension was then diluted 10-fold to stop the reaction, and then further diluted with the deionized water and centrifuged for three cycles. The sample was placed inside dialysis membrane tubes and dialyzed against slow-running deionized water for 2 to 4 days until the pH of supernatant was neutral. Subsequently, the colloidal solution was dispersed by ultrasound treatment in an ultrasonic cell crusher (X-1200D, ATPIO, China) for 25 min. Finally, CNC suspension was condensed using a rotary evaporator (RE-501A, ATPIO, China) until the concentration of CNC reached 2.0 wt.%.

Reviewer 3 Report

Comments and Suggestions for Authors

The manuscript is very well prepared, well illustrated, and contains interesting results. I have almost no comments on the content of the manuscript, but I still have some questions regarding the originality and novelty of the approach.

1) Lines 60-61. Is there not a single approach described in the literature to reduce the pitch of chiral nematics? If there are no such publications, then the work of the authors is fantastically new and it is necessary to emphasize this. Otherwise, it is necessary to cite existing works.

2) In addition, the idea of ​​modifying the supramolecular and interparticle structure of cellulose materials with the help of polyvalent metals is not new at all. But the authors indicated only an example of the introduction of sodium ions (line 54). In this regard, the Introduction section also needs improvement.

3) Section 2.1. I saw no point in transferring the material from this section to auxiliary materials. For what?

4) Section 2.2. On lines 76-77, the authors write that they prepared cellulose nanocrystals using a previously described method (ref. 26). As far as I understand, the cited work describes the production of nano-sized fibrous structures (i.e. CNF) from microfibrillar cellulose. At the same time, the authors write in the accompanying materials that they used microcrystalline cellulose as a starting material. I think that here it is necessary to more clearly describe the method of obtaining nanocrystalline cellulose (CNC), which the authors used in this work, because CNC and CNF are not the same thing, obviously.

Author Response

Comments and Suggestions 3:

The manuscript is very well prepared, well illustrated, and contains interesting results. I have almost no comments on the content of the manuscript, but I still have some questions regarding the originality and novelty of the approach.

1) Lines 60-61. Is there not a single approach described in the literature to reduce the pitch of chiral nematics? If there are no such publications, then the work of the authors is fantastically new and it is necessary to emphasize this. Otherwise, it is necessary to cite existing works.

Response: We regret for the description, and we revised it. “Limited studies have reported the modification methods for a decreased pitch [21, 27], which is of importance for chiral nematic materials”.

2) In addition, the idea of ​​modifying the supramolecular and interparticle structure of cellulose materials with the help of polyvalent metals is not new at all. But the authors indicated only an example of the introduction of sodium ions (line 54). In this regard, the Introduction section also needs improvement.

Response: We have added examples of modification of the molecular structure of cellulose by Cu2+ and Al3+. The corresponding references are as follows:

[16] Dai, S.D.; Prempeh, N.; Liu, D.G.; Fan, Y.M.; Gu, M.Y.; Chang, Y.. Cholesteric film of Cu(II)-doped cellulose colorimetric sensing of ammonia gas. Carbohydr Polym, 2017, 174, 531-539.

[22] Chen, C.; Sun, W.J.; Wang, L.; Tajvidi, M.; Wang, J.W.; Gardner, D.J. Transparent Multifunctional Cellulose Nanocrystal Films Prepared Using Trivalent Metal Ion Exchange for Food Packaging. ACS Sustainable Chem. Eng. 2022, 10, 29, 9419–9430.

3) Section 2.1. I saw no point in transferring the material from this section to auxiliary materials. For what?

Response: Thanks for all valuable suggestion. In section 2.1 the raw materials for film preparation have been described in detail in our previous studies, and we have placed the main raw materials in the main text.

4) Section 2.2. On lines 76-77, the authors write that they prepared cellulose nanocrystals using a previously described method (ref. 26). As far as I understand, the cited work describes the production of nano-sized fibrous structures (i.e. CNF) from microfibrillar cellulose. At the same time, the authors write in the accompanying materials that they used microcrystalline cellulose as a starting material. I think that here it is necessary to more clearly describe the method of obtaining nanocrystalline cellulose (CNC), which the authors used in this work, because CNC and CNF are not the same thing, obviously.

Response: We regret for the description, and we revised it. CNC colloid was fabricated according to our previous work [20]: 15 g of MCC was hydrolyzed for 50 min for preparation of the suspension. Hydrolysis was performed with 64 wt. % sulfuric acid at acid-to-pulp ratio of 8:1 at 45 °C and 60 °C. The suspension was then diluted 10-fold to stop the reaction, and then further diluted with the deionized water and centrifuged for three cycles. The sample was placed inside dialysis membrane tubes and dialyzed against slow-running deionized water for 2 to 4 days until the pH of supernatant was neutral. Subsequently, the colloidal solution was dispersed by ultrasound treatment in an ultrasonic cell crusher (X-1200D, ATPIO, China) for 25 min. Finally, CNC suspension was condensed using a rotary evaporator (RE-501A, ATPIO, China) until the concentration of CNC reached 2.0 wt.%.

[20] Zhao, G.M.; Zhang, S.; Zhai, S.; Pan, M.Z. Fabrication and characterization of photonic cellulose nanocrystal films with structural colors covering full visible light. J. Mater. Sci, 2020, 55(20): 8756-8767.

Round 2

Reviewer 1 Report

Comments and Suggestions for Authors

The authors should write the article title of Ref [31].

Comments on the Quality of English Language

The authors should write the article title of Ref [31].